# Integrated Proteomic and Metabolomic Analysis of Muscle Atrophy Induced by Hindlimb Unloading

**DOI:** 10.3390/biom15010014

**Published:** 2024-12-26

**Authors:** Yuan Wang, Xi Li, Na Li, Jiawei Du, Xiaodong Qin, Xiqing Sun, Yongchun Wang, Chengfei Li

**Affiliations:** 1Department of Aerospace Medical Training, School of Aerospace Medicine, Fourth Military Medical University, Xi’an 710032, China; wangcircle22@fmmu.edu.cn (Y.W.); lixi1908@fmmu.edu.cn (X.L.); 2019112026@bsu.edu.cn (N.L.); sunxiqing@fmmu.edu.cn (X.S.); 2Key Laboratory of Sports and Physical Fitness of the Ministry of Education, Beijing Sport University, Beijing 100084, China; jw_du@bsu.edu.cn

**Keywords:** muscle atrophy, hindlimb unloading, proteomics, metabolomics, integrated analysis

## Abstract

Skeletal muscle atrophy, which is induced by factors such as disuse, spaceflight, certain medications, neurological disorders, and malnutrition, is a global health issue that lacks effective treatment. Hindlimb unloading is a commonly used model of muscle atrophy. However, the underlying mechanism of muscle atrophy induced by hindlimb unloading remains unclear, particularly from the perspective of the myocyte proteome and metabolism. We first used mass spectrometry for proteomic sequencing and untargeted metabolomics to analyze soleus muscle changes in rats with hindlimb unloading. The study found 1052 proteins and 377 metabolites (with the MS2 name) that were differentially expressed between the hindlimb unloading group and the control group. Proteins like ACTN3, MYH4, MYBPC2, and MYOZ1, typically found in fast-twitch muscles, were upregulated, along with metabolism-related proteins GLUL, GSTM4, and NDUFS4. Metabolites arachidylcarnitine and 7,8-dihydrobiopterin, as well as pathways like histidine, taurine, and hypotaurine metabolism, were linked to muscle atrophy. Protein and metabolism joint analyses revealed that some pathways, such as glutathione metabolism, ferroptosis, and lysosome pathways, were likely to be involved in soleus atrophy. In this study, we have applied integrated deep proteomic and metabolomic analyses. The upregulation of proteins that are expressed in fast-twitch fibers indicates the conversion of slow-twitch fibers to fast-twitch fibers under hindlimb unloading. In addition, some differentially abundant metabolites and pathways revealed the important role of metabolism in muscle atrophy of the soleus. As shown in the graphical abstract, our study provides insights into the pathogenesis and treatment of muscle atrophy that results from unloading by integrating proteomics and metabolomics of the soleus muscles.

## 1. Introduction

Skeletal muscle atrophy caused by mechanical unloading leads to a decline in physical strength and an increased risk of disability [1]. Understanding these underlying processes is essential for developing strategies to combat this type of muscle deterioration. The disuse and unloading of muscles are likely the most significant factors that mediate spaceflight-induced muscle atrophy and have been extensively studied and reviewed. Skeletal muscle is a multicellular tissue consisting of myofibers, which account for approximately 40% of the overall weight of the human body and are key to exercise and energy metabolism [2]. Depending on the expression of myosin heavy-chain isoform as well as the metabolic and contractile properties, skeletal muscle fibers are classified into the following three main types: type I (slow oxidative), type IIa (fast oxidative), and type IIb (fast glycolytic) [3]. Unloading leads to a distinctive decline in the mass of the antigravity muscles, especially the soleus, which are abundant in slow-twitch type I fibers, as confirmed in human bed rest studies and tail-suspension rat models [4]. Proteomic studies related to muscle atrophy during mechanical unloading due to spaceflight or bed rest have been conducted [5,6]. However, single omics can hardly provide a comprehensive biological perspective.

Apart from the reduction in muscle mass and volume, muscle atrophy also involves complex metabolic changes within muscle cells, such as an imbalance between protein synthesis and degradation, and alterations in metabolic pathways [3]. Previous studies have found that the metabolic properties of muscle fibers are modified during unloading. After 3 weeks of unloading, the muscles of rats show a shift in fuel uptake from fatty acids to glucose [7]. Furthermore, studies have indicated an increase in triglycerides in atrophic muscles, along with changes in phospholipids during atrophy [8]. Although a number of metabolite levels have been shown to be altered in unloading-induced skeletal muscle atrophy, the application of metabolomics and proteomics joint analyses to investigate these mechanisms is rare.

To the best of our knowledge, this is the first study to combine proteomics and metabolomics to investigate soleus muscle atrophy induced by unloading. For this purpose, we used hindlimb unloading models in adult rats for up to 28 days to achieve an unloading effect. Analyzing changes in both proteins and metabolites simultaneously can help identify key biomarkers more accurately in the future, understand the molecular mechanisms underlying muscle atrophy, and potentially discover new therapeutic targets.

## 2. Materials and Methods

### 2.1. Animals

Male Sprague-Dawley (250–350 g) rats were sourced from the Experimental Animal Facility at the Air Force Medical University in China. Each animal was provided with a separate living space. Throughout the experiments, they had unrestricted access to both food and water. The environmental temperature was consistently kept within a range of 20 to 24 degrees Celsius, and a 12-h light/dark cycle was followed, with the lights turning on at 6:00 AM. Prior to commencing any experimental procedures, the rats were given a one-week period to adjust and become familiar with their new environment. Conducted ethically, all procedures involving animals adhered to the protocols approved by the Institutional Animal Care and Use Committee of the same institution (Approval code: 20210325). The specific number of animals utilized for each study is illustrated in the corresponding figure legends. The animals were assigned to two groups: the control group (CON) and the hindlimb unloading group (HU). The hindlimb unloading model is well established for simulating microgravity-induced cephalad fluid shift and cardiovascular deconditioning [9]. Two-month-old Sprague-Dawley rats in the HU group were subjected to a sustained −30° head-down tilt with their hindlimbs unloaded for 28 days to simulate the effects of microgravity. The CON group received identical handling, except for tail suspension.

### 2.2. Soleus Isolation

After administering pentobarbital sodium intraperitoneally at a dosage of 50 mg/kg, the animals were euthanized via abdominal aortic exsanguination. Then, the rats were positioned in a supine orientation for surgical procedures. The bilateral soleus muscles were meticulously excised, rinsed exhaustively with phosphate-buffered saline to remove residual blood and debris, and subsequently weighed. The right soleus muscle was promptly immersed in liquid nitrogen for freezing and then transferred to a −80 °C freezer after 15 min; the left soleus muscle was immersed in 10% neutral formalin for subsequent embedding.

### 2.3. Histological Analysis

Following paraffin embedding, the tissue specimens were sectioned into thin microtome slices. Subsequently, the muscle sections were subjected to haematoxylin–eosin staining (HE) staining. In preparation for immunofluorescence analysis, the muscle sections were first blocked with 5% Bovine Serum Albumin solution for one hour. Then, the sections were incubated with primary antibodies targeting Fast Myosin Skeletal Heavy chain (MYH1), Slow Skeletal Myosin Heavy chain (MYH7), Alpha-actinin 3 (ACTN3), Glutamine Synthetase (GLUL), Glutathione S-transferase M4 (GSTM4), Fructose-1,6-bisphosphatase isozyme 2 (FBP2), Yes-associated protein 1 (YAP1), and NADH dehydrogenase iron-sulfur protein 4 (NDUFS4) overnight at 4 °C. The MYH1 and MYH7 antibodies were procured from Servicebio, a Chinese company, and were utilized at a dilution ratio of 1:800. The remaining primary antibodies, sourced from Proteintech, another Chinese company, were applied at a dilution ratio of 1:200. Subsequently, the secondary antibodies were incubated for a duration of one hour at ambient temperature. The images were captured using a CaseViewer 2.4 imaging system.

### 2.4. Proteomics Analysis

Proteins from muscle tissue samples were extracted from both the control and HU groups. Utilizing RIPA-lysis buffer, the extraction was executed on a cohort of three rats per group. The samples adhered to the filter-aided sample preparation methodology as per the manufacturer’s guidelines. Proteins underwent enzymatic digestion with trypsin, applied at a ratio of 50:1 to protein. Centrifugation was used to collect peptide samples for proteomic analysis, followed by reconstituted peptides in a sample loading buffer. Following that, we conducted an LC−MS/MS Analysis. The peptides were solubilized in solvent A and directly loaded onto a bespoke reversed-phase analytical column (25 cm × 100 μm i.d.). The chromatographic separation employed a binary solvent system: solvent A (0.1% formic acid, 2% acetonitrile in water) and solvent B (0.1% formic acid, 90% acetonitrile in water). A gradient elution was applied (0–22.5 min, 6–22% B; 22.5–26.5 min, 22–34% B; 26.5–28.5 min, 34–80% B; 28.5–30 min, 80% B) at a flow rate of 700 nL/min on an EASY-nLC 1200 UPLC system. The peptides were then subjected to nano-electrospray ionization, followed by analysis using an Orbitrap Exploris 480 mass spectrometer. The FAIMS CV was set to 45 V, and the electrospray voltage was 2300 V. The Orbitrap was configured to record precursor ions and their fragments within a 350–1400 *m*/*z* range at a resolution of 60,000 for full MS scans and 15,000 for MS/MS scans, with the HCD NCE set at 27%. The automatic gain control target was fixed at 1 × 10^6^, and the maximum injection time was set to 22 ms. Data Independent Acquisition (DIA) data were analyzed utilizing the DIA-NN algorithm, version 1.8. The tandem mass spectrometry data were processed against a database comprising Rattus norvegicus_10116_PR_20230103.fasta, which contained 47,945 entries, complemented by a reversed decoy database. The search parameters were configured with trypsin/P as the enzyme for peptide cleavage, permitting a maximum of one missed cleavage. Static modifications included excision of the N-terminal methionine and carbamidomethylation of cysteines. The False Discovery Rate (FDR) was adjusted to <1%. PPI analysis was performed using the STRING database (https://string-db.org, version 12.0; 26 July 2023) and Cytoscape software (version 3.8.1).

### 2.5. Western Blot Analysis

The total tissue samples underwent separation via 10% SDS-PAGE gel electrophoresis and were then transferred onto a 0.45-μm polyvinylidene fluoride membrane. The membranes were blocked with 5% non-fat milk in Tris-buffered saline with Tween 20 for 2 h at room temperature. The membranes were then incubated with antibodies against ACTN3 (Proteintech, 1:2000, Wuhan, China), GLUL (Proteintech, 1:4000, Wuhan, China), GSTM4 (Proteintech, 1:2000, Wuhan, China), FBP2 (Proteintech, 1:2000, Wuhan, China), YAP1 (Proteintech, 1:5000, Wuhan, China), NDUFS4 (Proteintech, 1:2000, Wuhan, China), and GAPDH (Zhuangzhi, 1:2000, Xian, China) allowing for overnight incubation at 4 °C. GAPDH was used as a protein loading control. The secondary antibody (Zhuangzhi, 1:5000) was either an anti-mouse or anti-rabbit secondary antibody. Protein bands were visualized by enhanced chemiluminescence (Millipore, Burlington, MA, USA). The band intensities were analyzed by ImageJ (https://imagej.nih.gov/ij/index.html, accessed on 26 July 2023).

### 2.6. Metabolomics Analysis

Muscle samples were stored in a −80 °C low-temperature freezer. The samples were transferred to EP tubes and an extraction solution containing isotopically labeled internal standards was added. After vortex mixing for 30 s, followed by sonication, the samples were left to stand at −40 °C for 1 h. After centrifugation for 15 min, the supernatant was collected and placed in injection vials for machine detection. An equal amount of supernatant from all samples was mixed to form a quality control sample, which was also analyzed on the machine. In this project, a Vanquish (Thermo Fisher Scientific, Waltham, MA, USA) ultra-high-performance liquid chromatograph was used, with a Waters ACQUITY UPLC BEH Amide liquid chromatography column employed for the chromatographic separation of target compounds. The liquid chromatography mobile phase A was the aqueous phase, and phase B was acetonitrile. The sample tray temperature was set to 4 °C, with an injection volume of 2 μL. The Orbitrap Exploris 120 mass spectrometer was used to collect both first-order and second-order mass spectrometry data under the control of the Xcalibur 4.4 software (Thermo, Waltham, MA, USA). Detailed parameters are as follows: Sheath gas flow rate: 50 Arb, Aux gas flow rate: 15 Arb, Capillary temperature: 320 °C, Spray Voltage: 3.8 kV for positive mode or −3.4 kV for negative mode. The raw data, after being converted to mzXML format using ProteoWizard software (V3.0.8789), were processed for metabolite identification using a custom R package, with the BiotreeDB (V3.0) database utilized for this purpose. Subsequently, a self-developed R package was used for visualization analysis.

### 2.7. Statistical Analysis

Data analysis was conducted using GraphPad Prism software (8.3.0.). The results are presented as the mean ± *SEM* of three independent experiments. Comparisons were analyzed using an unpaired *t*-test. A *p*-value < 0.05 is considered as statistically significant.

## 3. Results

This section may be divided into subheadings. It should provide a concise and precise description of the experimental results, their interpretation, and the experimental conclusions that can be drawn.

### 3.1. Soleus Muscle Atrophy Induced by Hindlimb Unloading

The hindlimb unloading rat model is a typical model commonly used for weightlessness-induced muscle atrophy (Figure 1A). In this study, we found that HU for 28 days did not affect adult rat body mass (Figure 1B), which is similar to the results of previous studies [10]. Nevertheless, the soleus muscles of rats showed decreases in all dimensions in the HU group in comparison with the CON group. As a result of unloading, soleus mass (Figure 1C) and soleus/body mass (Figure 1D) were lower in the HU group compared to the control group. According to HE staining, the mean fiber cross-section area of the soleus was smaller after HU (Figure 1E). In conclusion, HU could induce soleus atrophy in rats exposed to hindlimb weightlessness for 28 days.

### 3.2. Proteomic Changes in Soleus of Rats After HU

We used LC–MS/MS analysis to analyze soleus tissue samples from the CON group and the HU group, which allowed us to comprehensively elucidate the proteomic regulation of muscle atrophy. In order to evaluate the repeatability of muscle samples, principal component analysis (PCA) was performed on all samples. According to PCA, the samples were clustered mainly by tissue of origin (Figure 2A). After the process of data filtering, a total of 4668 proteins were successfully identified, out of which 4495 proteins were quantified for the purpose of comparison (Appendix A). We set the threshold for significant upregulation to be fold change (FC) > 1.5 (*p* < 0.05) and the threshold for significant downregulation to be fold change < 1/1.5 (*p* < 0.05). As a result, 1052 DEPs were found in the HU group compared to the CON group, with 787 upregulated and 265 downregulated (Appendix A). The heatmap showed the clustering relationship of the relative expression of differentially expressed proteins (DEPs) (Figure 2B). Subcellular localization analysis revealed that the largest number of entities were cytoplasmic, followed by nuclear and extracellular (Figure 2C).

Gene Ontology (GO) enrichment analysis (Figure 2D) and KEGG analysis (Appendix A) for DEPs were performed. A total of 45 GO categories were classified, with DEPs being enriched in biological processes, cellular components, and molecular functions. The majority of enriched categories in the cellular components were intracellular anatomical structures and cytoplasm. Apparently, the vast majority of biological process terms were relevant to metabolic processes, such as organic substance metabolic processes, cellular metabolic processes, and primary metabolic processes (Figure 2D), suggesting that these processes played an important role in soleus muscle atrophy induced by unloading. For molecular function, 15 GO terms were enriched, among which protein binding and ion binding were dominant. KEGG analysis revealed that vitamin digestion and absorption, complement and coagulation cascades, and cholesterol metabolism were enriched pathways. In addition, DEPs were also enriched in glycolysis/gluconeogenesis and protein digestion and absorption, according to KEGG pathway analysis. This result was consistent with previous reports that glucose oxidation was increased in unloaded muscle [5] and the notion that the atrophic state is attributed to enhanced proteolysis and compromised protein synthesis. Then, the PPI network of the top 25 upregulated and top 25 downregulated DEPs was constructed to filter out the key genes involved in muscle atrophy (Figure 2F). Eight hub genes, including Ankyrin Repeat Domain 2 (ANKRD2), Myosin Light Chain 2(MYL2), ATPase sarcoplasmic/endoplasmic reticulum Ca2+ transporting 1 (ATP2A1), Titin (TTN), Myosin Binding Protein C2(MYBPC2), ACTN3, Myosin Heavy chain 4(MYH4), Myosin Heavy chain 3(MYH3), were acquired by taking the overlap among maximum cluster centrality MCC (Maximal Clique Centrality), MNC (Maximum Neighborhood Component) and Degree.

### 3.3. Proteomic Analysis Revealing DEPs Related to Unloading Muscle Atrophy

Proteins were selected for further analysis based on both their high fold change and probable role in muscle atrophy, according to the proteomic data (Figure 3A). The increased expression of MYH4, ACTN3, myozenin 1 (MYOZ1), a structural protein involved in muscle regeneration [11], and MYBPC2 indicated an increased proportion of fast-twitch fibers in the unloaded soleus because these four proteins are mainly expressed in fast-twitch muscle fibers [12,13]. This result is consistent with previous studies showing that gravitational unloading leads to the transformation of slow-twitch muscle fibers into fast-twitch fibers [14]. Similarly, ATP2A1, which is also known as the sarco (endo) plasmic reticulum Ca2+ ATPase type 1 (SERCA1) and is expressed in fast-twitch muscles [15], was upregulated in the unloaded soleus. The decreased expression of myozenin 2 (MYOZ2), which is specifically expressed in slow skeletal muscle fibers [16], further supports the shift from slow-twitch muscle fibers to fast-twitch fibers. Additionally, the muscle contraction-related troponin family member troponin C2 (TNNC2), which is expressed in fast-twitch fibers [17], was upregulated, whereas Troponin I1 (TNNI1) and Troponin C1 (TNNC1), which are expressed in slow skeletal muscle fibers [18,19], were downregulated in the soleus of unloading rats. Figure 3B shows the facilitative effect of hindlimb unloading on the transformation of slow-twitch muscle fiber types to fast-twitch fibers in the soleus muscle. This transformation is characterized by an enhancement in the percentage of fast-twitch fibers and a concomitant reduction in the percentage of slow-twitch fibers subsequent to hindlimb unloading. In addition, we conducted both Western blot (WB) and immunofluorescence assays to validate the proteomic findings of ACTN3 (Figure 3C). The results from these analyses revealed a significant upregulation in the expression levels of ACTN3 following hindlimb unloading. This finding is consistent with the role of ACTN3 in the transformation of muscle fiber types, where it may contribute to the shift from slow-twitch to fast-twitch fibers in the soleus muscle [20]. The decreased expression of YAP1, a key effector of the Hippo pathway and a key regulatory factor involved in skeletal muscle development and regeneration [21], is shown in this study (Figure 3A). It was also validated by WB and immunofluorescence assays (Figure 3D). These observations suggest that the unloading protocol may induce a shift in muscle fiber composition, potentially altering muscle function and performance.

In addition to muscle contraction-related structural proteins, it is worth noting that several proteins involved in metabolic processes were dysregulated in the unloaded soleus of rats. The expression of GLUL, the only enzyme capable of synthesizing glutamine, increased after hindlimb unloading. According to our proteomics analysis, the glutathione transferase family members A3, P1, T3, M1, M3, and M4 (GSTA3, GSTP1, GSTT3, GSTM1, GSTM3, and GSTM4), which are located in the cytoplasm of both cardiac and skeletal muscles [22], were upregulated. FBP2 is a muscle-specific form of fructose bisphosphatase that converts fructose-1,6-bisphosphate to fructose-6-phosphate in mammalian muscle and is abundant in fast-twitch muscle tissues. Similar to previous findings in mice [23], the expression of FBP2 increased in response to gravity unloading in the soleus muscle of rats. NDUFS4 plays a crucial role in mitochondrial energy metabolism and is also downregulated after hindlimb unloading. Additionally, the protein expression of several other NADH dehydrogenases, including NADH:ubiquinone oxidoreductase subunit S6-like 1(NDUFS6L1) and NADH:ubiquinone Oxidoreductase Subunit A7(NDUFA7), was also decreased after hindlimb unloading (Figure 4A). Consistent with our data from proteomics analysis, western blot results indicated that the expression of ACTN3, GLUL, GSTM4, and FBP2 was upregulated, and the protein levels of NDUFS4 were decreased in unloading muscles (Figure 4B). The expression of GLUL, GSTM4, FBP2, and NDUFS4 was also verified by immunofluorescence (Figure 4C). Based on these changes in representative proteins, we can gain a better understanding of disuse muscle atrophy induced by hindlimb unloading.

### 3.4. Metabolic Analysis Revealing Metabolites Changes of Soleus in Response to Unloading

Because metabolites such as glucose, lipids, and triglycerides are potential modulators of muscle atrophy [24,25], we used LC−MS/MS metabolomic analysis to investigate the role of metabolites in the soleus after unloading. Before conducting a detailed analysis of specific metabolic alterations, we applied the OPLS-DA model to preliminarily evaluate metabolic differences between the HU group and the CON group (Figure 5A). The permutation histogram test of the OPLS-DA model showed a clear separation between the HU group and the CON group, suggesting reproducible chemical differences.

Following a preliminary observation of the disparities, we employed both univariate and multivariate statistical methods to identify distinct metabolites in the soleus. Based on the screening criteria *p* < 0.05, Variable Importance in the Projection (VIP) > 1, and FC > 1.5 or FC < 0.67 in this study, a total of 4285 differentially abundant metabolites (DAMs) were identified, of which 1754 DAMs were upregulated, and 2531 DAMs were downregulated. Among these DAMs, 377 had a member signature (MS2) name provided by the mass spectrometry qualitative matching analysis (Figure 5B). According to the chemical classification information, DAMs were classified into 23 types, and the proportions of each type of metabolite are shown (Figure 5C). Among the 377 metabolites, organ heterocyclic compounds accounted for the highest proportion, accounting for 17.37%, followed by lipid and lipids-like molecules, accounting for 15.7%. The top 20 most significantly changed metabolites are displayed in Figure 5D, with these metabolites primarily concentrated in the pathways of amino acid metabolism and lipid metabolism. The top 5 upregulated metabolites were arachidyl carnitine, 7,8-dihydrobiopterin, asn-gly-asn, thr-leu, and ser-leu, and the top 5 downregulated metabolites were diallyl trisulfide, 5-(4-morpholinylsulfonyl)-1h-indole-2,3-dione, phenamil, 5-[3-(trifluoromethyl)phenyl] furfural and epigallocatechin. Then, we performed KEGG enrichment analysis on the DAMs to better understand the metabolic pathways (Figure 5E). According to KEGG analysis, 108 metabolic pathways were classified and annotated. The top 10 significantly enriched KEGG metabolic pathways according to the differential abundance score were biosynthesis of amino acids, glycerophospholipid metabolism, d-amino acid metabolism, ABC transporters, histidine metabolism, central carbon metabolism in cancer, alanine, aspartate, and glutamate metabolism, protein digestion, absorption, aminoacyl-tRNA biosynthesis, and choline metabolism in cancer (Figure 5E). Notably, the protein digestion and absorption pathways were co-enriched in proteomics and metabolomics, suggesting a pivotal role for this pathway in muscle atrophy induced by unloading. Additionally, pathway analysis by MetaboAnalyst was performed. The results showed that these DAMs were significantly enriched in pathways including alanine, aspartate, and glutamate metabolism; D-glutamine and D-glutamate metabolism; arginine and proline metabolism; histidine metabolism; and taurine and hypotaurine metabolism (Figure 5F). It is noteworthy that histidine metabolism and alanine, aspartate, and glutamate metabolism were enriched in both KEGG analysis and pathway analysis by MetaboAnalyst, indicating their important role in soleus atrophy.

### 3.5. Integration Analysis of Proteomics and Metabolomics

In order to clarify the mechanism contributing to soleus atrophy caused by HU, we performed an integrated proteomic-metabolomic analysis. There were 45 and 108 KEGG pathways specifically enriched in proteomics and metabolomics, respectively (Figure 6A,B), from which we screened 16 common pathways, including purine metabolism, protein digestion and absorption, lysosome, glucagon signaling pathway, glutathione metabolism, and ferroptosis. Figure 6C,D show the KEGG pathway maps of lysosomes and ferroptosis. Some lysosomal acid hydrolases and lysosomal membrane proteins were upregulated, indicating the enhanced hydrolytic function of lysosomes, which may be responsible for muscle atrophy. According to the pathway map of ferroptosis, the amount of glutathione disulfide (GSSG) decreased, which may be due to the decreased activity of GPX4. In addition, the upregulation of FERRITIN, STEAP3, and NOX2 indicated the possibility of iron overload and oxidative stress occurrence. The glucagon signaling pathway map showed decreased amounts of malate and 2-oxoglutarate in the citrate cycle and increased expression of LDH. (Appendix A).

Then, we conducted a correlation analysis between the six DEPs that we verified and all DAMs between HU and CON (Figure 6E). The metabolites that correlated with these six proteins contained lipid-like molecules, amino acids, and intermediates of the tricarboxylic acid cycle. Notably, GLUL, a glutamine synthetase, exhibited a significant negative correlation with glutamine. ACTN3, FBP2, and GSTM4 were positively correlated with some dipeptides, such as Ile-Leu and Hly-His. A significant positive correlation was observed between NDUFS4 and malic acid. The majority of metabolites that were significantly negatively correlated with GSTM4 were amino acids. YAP1 exhibited a significant positive correlation with tiglylcarnitine, a short-chain acylcarnitine.

## 4. Discussion

Skeletal muscle atrophy induced by spaceflight leads to functional deficits, such as a decrease in maximum force generation, increased fatigability, and higher activation threshold, affecting the physical function and work efficiency of astronauts. Even short-term spaceflight can result in significant muscle atrophy, with the most significant atrophy in antigravity muscles, such as the soleus, even when exercise countermeasures are taken. Muscle atrophy induced by spaceflight and muscle atrophy due to aging or disuse from injury exhibit similar manifestations and may share some common underlying mechanisms. It has been shown that skeletal muscle atrophy under spaceflight conditions is accompanied by metabolic changes, including deficiency in vitamin D and decreased citrate synthesis in muscle tissue [26]. This study conducted a joint analysis of proteomics and metabolomics for the first time on the atrophy of the soleus muscle induced by hindlimb unloading. The present findings contribute to a better understanding of the spatiotemporal molecular adaptation of skeletal muscles to spaceflight and provide a large-scale database for the design of effective countermeasure protocols.

We first focused on the DEPs of the soleus muscle in the CON and HU groups. A series of proteins that are expressed in fast muscle fibers showed significant upregulation in their levels, such as MYH4, ACTN3, MYOZ1, myosin binding protein C (MYBPC2), ATP2A1, and TNNC2. According to a previous study, the accumulation of ACTN3 alters type II glycolytic myofibers, subsequently accelerating age-associated muscle atrophy. Furthermore, reduction in ACTN3 expression ameliorated muscle atrophy [27]. In our study, we speculated that the upregulation of ACTN3 might be the key to muscle atrophy induced by unloading. Meanwhile, some proteins that were exclusively expressed in slow muscle fibers were significantly downregulated, such as TNNI1, TNNC1, and MYOZ2. In addition, some differentially expressed metabolic proteins have also been confirmed in previous studies, which indicate that simulated weightlessness causes a transition from slow muscle fibers to fast muscle fibers [14], thereby reducing muscle endurance. The NADH dehydrogenases NDUFS4, NDUFS6L1, and NDUFA7, which play a key role in the process of oxidative phosphorylation, were downregulated. The downregulation of these proteins after unloading is consistent with previous findings of compromised mitochondrial respiration in glucocorticoid-induced muscle atrophy [28]. FBP2, which has been reported to inhibit mitochondrial biogenesis [29], was upregulated in our study. This result is consistent with the metabolic characteristics of suppressed mitochondrial synthesis and oxidative phosphorylation in fast muscle fibers [30]. Additionally, mitochondrial dysfunction has been implicated in the shift of muscle fibers from an oxidative type (slow-twitch muscle fibers) to a glycolytic type (fast-twitch muscle fibers) type [31,32]. Our results indicate that FBP2 may be a key molecule in mediating mitochondria function during the transition from slow muscle to fast muscle. However, the muscle proteome analysis of Facioscapulohumeral dystrophy patients reveals decreased FBP2, whereas the upregulation of FBP2 was found in our study, indicating the distinct molecular mechanisms underlying muscle atrophy in various muscle atrophy models [33]. As a conditional essential amino acid during catabolic states, glutamine prevents glucocorticoid-induced muscle atrophy in rats [34]. We speculated that the increased expression of glutamine synthetase GLUL may be a compensation and negative feedback response to muscle atrophy. In addition, YAP1 has emerged as an important player in mechanotransduction, transmitting mechanical cues into a transcriptional cell response [21]. Although a previous study found the pivotal role of YAP1 in regulating skeletal muscle size [35], there is no research on the change in the protein level of YAP1 in response to skeletal muscle unloading. Our study showed for the first time that YAP1 expression dramatically decreased after 28 days of hindlimb unloading. YAP abundance and activity in muscles are increased following injury or degeneration of motor nerves, which is a process to mitigate neurogenic muscle atrophy. The increased YAP1, possibly as a subsequent biological response, promotes protein synthesis in denervated skeletal muscles [35]. However, in our study, the downregulation of YAP1 after HU may be one of the initiating factors of disuse atrophy. The role of YAP1 in muscle atrophy caused by unloading remains to be further investigated.

The KEGG pathway enrichment analysis for DEPs has identified that the pathways for vitamin digestion and assimilation, the cascades of complement and coagulation, and the metabolism of cholesterol are notably enriched. Previous studies have reported that vitamin C enhances muscle repair by stimulating myoblast growth, suggesting a role for vitamin C in the regeneration of skeletal muscle [36]. It has been reported that complement component 1q activates the Wnt signaling pathway, thus leading to the progression of muscle fibrosis by promoting fibroblast proliferation and decreasing the proliferative capacity of satellite cells [37]. Furthermore, a previous study reported that proteins of the complement and coagulation cascades were affected by bed rest unloading according to the plasma proteome [38]. Our results provide direct evidence of the crucial role of complement and coagulation cascades in muscle atrophy. Additionally, muscle cholesterol metabolism plays an important role in the pathophysiology of Duchenne muscular dystrophy, which is a direct therapeutic target in Duchenne muscular dystrophy [39].

Through metabolomic studies, we can gain a deeper understanding of the changes in metabolites within muscle cells during atrophy, which can help reveal the molecular mechanisms of muscle atrophy and potentially identify new therapeutic targets. Among the top 20 most significantly changed metabolites, three upregulated dipeptides containing leucine may promote muscle growth and repair as a feedback mechanism of soleus atrophy because leucine, as an essential branched-chain amino acid, enhances skeletal muscle growth [40]. The increased amount of arachidyl carnitine indicated changes in the fatty acid oxidation process. The differential metabolite pathway enrichment analysis showed that histidine metabolism and alanine, aspartate, and glutamate metabolism were enriched in both KEGG and MetabolAnalyst pathway analyses. Histidine is an important substrate for the synthesis of carnosine, which is located mainly in skeletal muscles and plays a pivotal role in the muscle contraction process through calcium regulation and pH buffering [41]. Alanine, aspartate, and glutamate, which are derived from intermediates of the citric acid cycle, are closely related to the energy metabolism of the soleus. Alanine catabolism in the liver has been reported to promote skeletal muscle atrophy in type 2 diabetes [42]. Additionally, the taurine and hypotaurine metabolism pathways were enriched in the MetaboAnalyst analysis. Taurine and hypotaurine metabolism play a crucial role in modulating excitation–contraction coupling in skeletal muscles, and 70% of the total taurine in the body is stored in skeletal muscle. Knockout of the taurine transporter in mice results in low taurine concentrations in the muscle and is associated with myofiber necrosis and diminished exercise capacity. Taurine metabolism disorder is an important mechanism in Duchenne muscular dystrophy [43]. In our study, these metabolite pathways were first reported to be involved in unloading-induced muscle atrophy. In addition, the DAMs observed in our study share similarities with those found in the plasma of patients with sarcopenia, such as the elevation of carnitine and alterations in the histidine metabolism pathway [44].

The integration of proteomics and metabolomics has provided a deeper understanding of the molecular mechanisms underlying muscle atrophy. Through KEGG pathway analysis of DEPs and DAMs, we screened 16 common KEGG pathways in which differentially expressed proteins and metabolites between CON and HU were involved, including protein digestion and absorption, ferroptosis, lysosome, glutathione metabolism, and purine metabolism. The enrichment of protein digestion and absorption and the enhanced expression of some lysosomes imply that skeletal muscle atrophy occurs because of an imbalance in myofibrillar protein turnover, favoring protein degradation over protein synthesis. As a process of iron-dependent programmed cell death, the ferroptosis pathway was also enriched. It has been found that high iron content is associated with diminished muscle mass in the elderly and in aged rats [45]. Our study indicates iron overload and increased oxidative stress according to the KEGG pathway map and is the first to reveal that ferroptosis may be involved in disuse muscle atrophy. Additionally, the glucagon signaling pathway map shows increased expression of LDH and decreased amounts of malate and 2-oxoglutarate. This finding confirms the transition from slow-twitch to fast-twitch muscle fibers in the soleus exposed to unloading, because fast-twitch muscle fibers rely more on glycolysis than oxidative phosphorylation for energy supply [46]. Although GLUL is a glutamine synthetase, glutamine was found to be negatively correlated with GLUL. This result may be due to the decreased amount of glutamate according to the pathway map because glutamate is the substrate for the synthesis of glutamine. The positive correlation between YAP1 and tiglylcarnitine indicates a possible important role of YAP1 in metabolism. The role of YAP1 in metabolic pathways is being increasingly recognized. YAP1 is reported to be involved in arachidonic acid metabolism in the process of liver carcinogenesis [47]. Previous studies have also confirmed the important role of YAP1 in glycolysis [48]. Our results suggest that YAP1 may play a role in fatty acid oxidation in the soleus muscle. In addition, the amount of tiglylcarnitine, which is a short-chain acylcarnitine and is involved in the β-oxidation of fatty acids [49], decreased. This is consistent with the fact that the proportion of energy supply from fatty acid oxidation is reduced in fast muscle fibers compared with slow muscle fibers [46]. The positive correlation between NDUFS4 and malic acid is consistent with the decreased amount of malate and 2-oxoglutarate according to the above glucagon signaling pathway KEGG map, indicating decreased activity of the tricarboxylic acid cycle in the soleus muscle after unloading.

## 5. Conclusions

In this study, we conducted a combined analyses of proteomics and metabolomics. The analysis results confirm that unloading can lead to the transformation of slow-twitch fibers into fast-twitch fibers in the soleus muscle. Additionally, through the combined analysis of omics, we discovered some new molecules and metabolic pathways that may be involved in the process of atrophy of the soleus muscle caused by unloading, such as the YAP1 and ferroptosis signaling pathways. Of course, this study has some limitations, such as not verifying the levels of metabolites and not confirming changes in some pathways, such as the ferroptosis pathway and the enhancement of lysosomal enzyme activity.

## Figures and Tables

**Figure 1 biomolecules-15-00014-f001:**
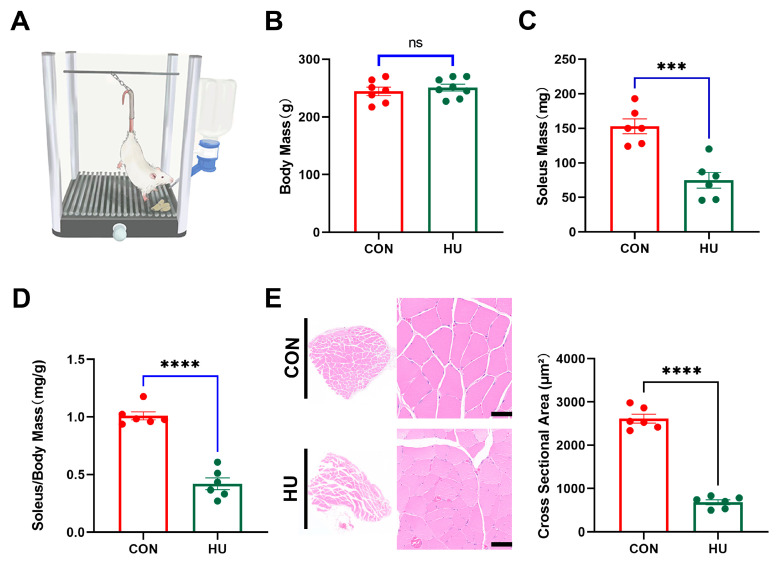
Analysis of soleus muscle atrophy induced by hindlimb unloading (n = 6). (**A**) the model of hindlimb unloading in rats. (**B**) Body mass collected in the CON group (CON) and hindlimb unloading (HU). (**C**) Wet soleus mass and Soleus/Body Mass (**D**). (**E**) Myofibre cross-sectional area (CSA) of soleus. (Scale Bar = 50 µm. The data represent the mean ± *SEM*, n = 3; *ns* stands no statistical significance, *** *p* < 0.001, **** *p* < 0.0001).

**Figure 2 biomolecules-15-00014-f002:**
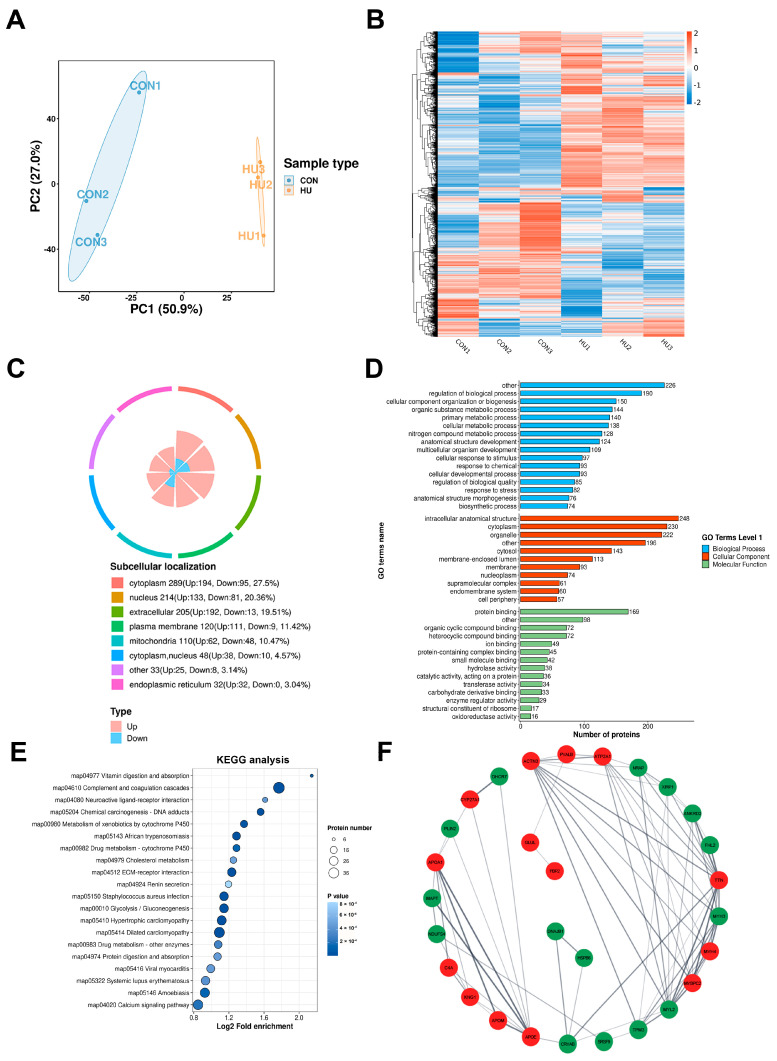
Proteomic analyses of the HU group and the CON group (n = 3). (**A**) PCA analysis of HU and CON groups. The horizontal and vertical axes show the degree of interpretation of PC1 and PC2, with higher values increasing the degree of interpretation. The degree of aggregation within groups represents the repeatability of the grouped samples, and the repeated samples in each group tend to be clustered together. (**B**) Heatmap of hierarchical clustering analysis for all groups. The horizontal coordinate represents the grouping of different samples. Red indicates the high expression of the substance, and blue indicates the low expression of the substance. (**C**) Pie plot showing the different classifications of DMPs. Different color blocks indicate different taxonomic categories, and percentages indicate the proportion of proteins belonging to that type among all identified proteins percentage of the number. (**D**) GO classification shows the DEPs classification. The horizontal axis represents the number of differentially expressed proteins in the classification, and the vertical axis represents the secondary functional classification in the primary classification of GO (Biological process/Cellular component/Molecular function). The figure shows the first-order classification of GO in different colors. (**E**) KEGG analysis of DEPs. The dotplot shows the 20 most significantly enriched functions. The vertical axis represents the KEGG function description information, while the horizontal axis represents the enrichment significance *p*-value of the-Log10 transformation; the higher the value, the stronger the enrichment significance. (**F**) Protein–protein interaction (PPI) network of DEGs between HU and CON based on the STRING database. Red nodes indicate upregulated proteins, and green nodes represent downregulated proteins.

**Figure 3 biomolecules-15-00014-f003:**
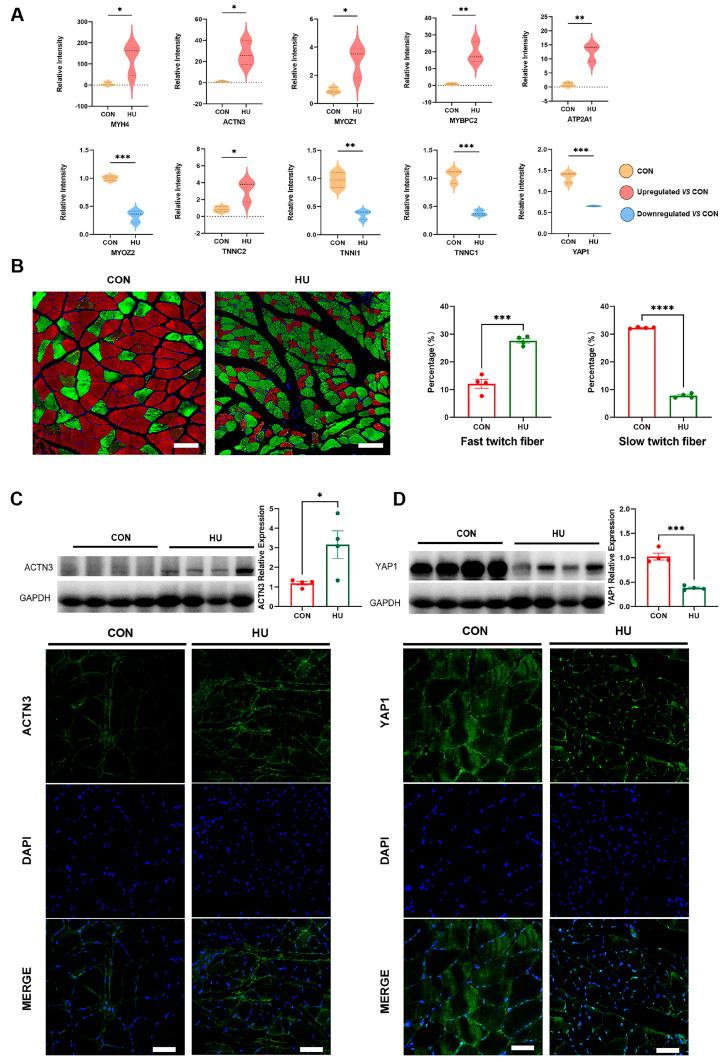
Effect of hindlimb unlaoding on muscle fiber type. (**A**) Violin plots showing DEPs related to myofiber type. (**B**) The immunofluorescence assays of fast-twitch fibers (green) and slow-twitch muscle fiber types (red). (**C**,**D**) Western blot and Immunofluorescent staining with target ACTN3 and YAP1. (Scale Bar = 50 µm. The data represent the mean ± *SEM*, n = 4; * *p* < 0.05, ** *p* < 0.01, *** *p* < 0.001, **** *p* < 0.0001).Western blot original images can be found in Appendix A.

**Figure 4 biomolecules-15-00014-f004:**
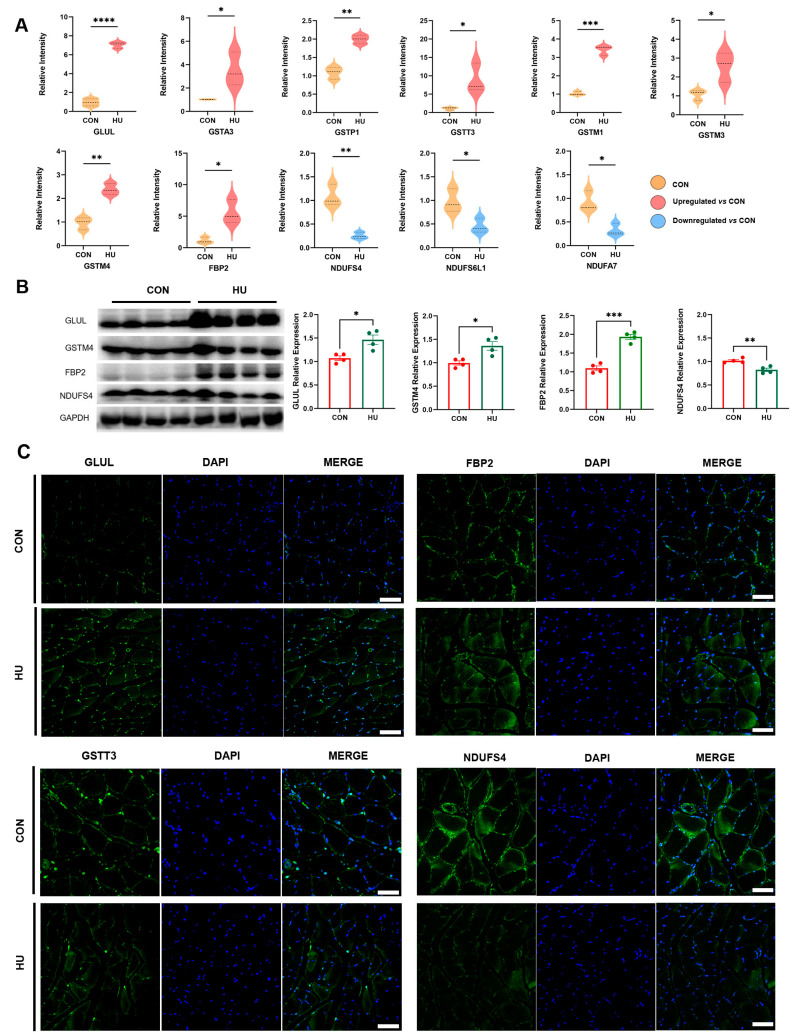
Validation of target DEPs. (**A**) Violin plots showing DEPs related to metabolic processes. (**B**) Western blot and analysis of target DEPs. (**C**) Immunofluorescent staining with target DEPs. (Scale Bar = 50 µm. The data represent the mean ± *SEM*, n = 4; * *p* < 0.05, ** *p* < 0.01, *** *p* < 0.001, **** *p* < 0.0001). Western blot original images can be found in Appendix A.

**Figure 5 biomolecules-15-00014-f005:**
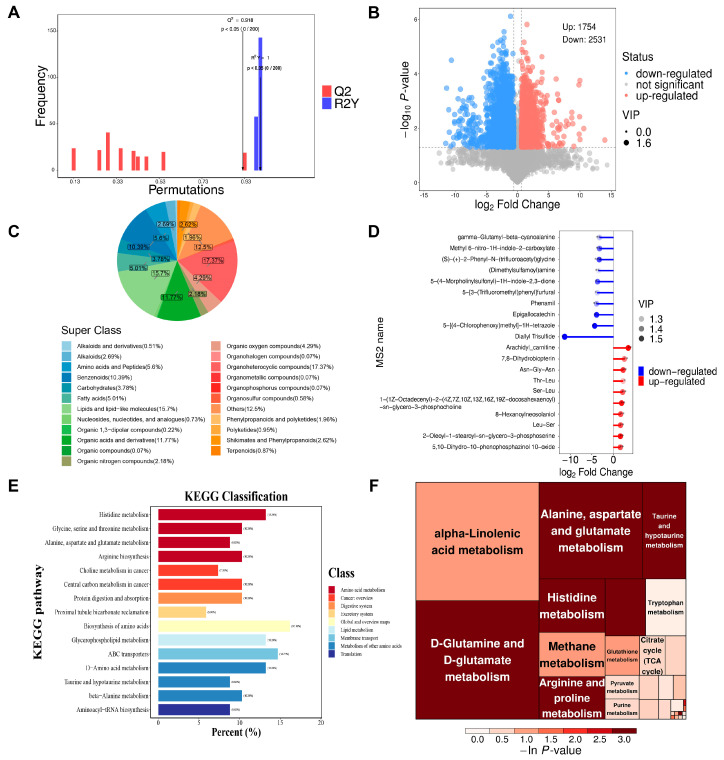
Metabolomic analysis of the HU group and the CON group (n = 3). (**A**) OPLS-DA model permutation plots for HU and CON groups showing a high explanatory and predictive power for categorical variables. (**B**) Volcano plots showing differential metabolites with the screening criteria including *p* < 0.05, VIP> 1, and FC > 1.5 or FC < 0.67. The red and blue dots represent metabolites that increased and decreased significantly in the HU group, respectively. (**C**) Pie plot showing the different classifications of DAMs. Different color blocks indicate different taxonomic categories, and percentages indicate the proportion of metabolites belonging to that type among all identified metabolites percentage of the number. (**D**) Matchstick analysis for DAMs between the CON and HU groups showing the top 10 upregulated and downregulated metabolites in terms of fold changes. The horizontal axis of the graph shows the logarithmically transformed fold changes, and the color intensity of the points represents the VIP values. (**E**) KEGG Classification of DAMs. The abscissa represents the percentage of the number of annotated differential metabolites under a certain pathway to the number of all annotated differential metabolites, and the ordinate represents the enriched KEGG metabolic pathway name. (**F**) Metabolic pathway analysis using MetaboAnalyst. In the tree map, each square represents a metabolic pathway. The size of the square indicates the impact factor of the pathway in the topological analysis. The larger the size, the greater the impact factor. The color of the square indicates the *p*-value of the enrichment analysis. The darker the color, the smaller the *p*-value and the more significant the enrichment.

**Figure 6 biomolecules-15-00014-f006:**
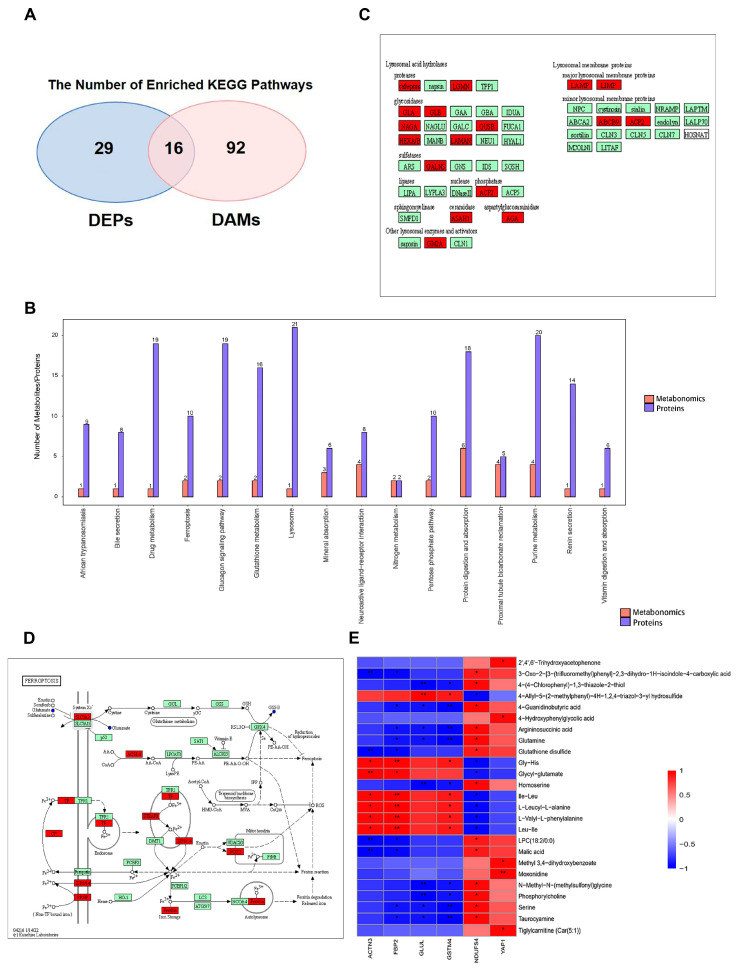
Integrated analyses of Proteomics and Metabolomics. (n = 3) (**A**) Venn diagrams showing the number of protein (blue) and metabolite (red) pathways significantly changed after hindlimb unloading, followed by a list of pathways that were regulated at both the protein and metabolite levels. (**B**) Barplot of KEGG pathway enrichment. (**C**,**D**) The KEGG pathway maps of lysosomes (**C**) and ferroptosis (**D**). (Blue indicates downregulation and red indicates upregulation. The small disc represents metabolites. The rectangle represents proteins.) (**E**) Heat map of correlation between DEPs ACTN3, FBP2, GLUL, GSTM4, NDUFS4, YAP1, and metabolites. (* *p* < 0.05, ** *p* < 0.01).

## Data Availability

The proteomics data, derived from mass spectrometry analysis, have been submitted to ProteomeXchange through the PRIDE partner repository and are accessible with the dataset identifier PXD057671. The data sets employed for analysis in the present study are accessible from the lead researcher upon request, which is deemed reasonable.

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
