# Peer review of "Integrated Proteomic and Metabolomic Analysis of Muscle Atrophy Induced by Hindlimb Unloading"

_biomolecules, 2024, doi:10.3390/biom15010014_

Round 1

Reviewer 1 Report

Comments and Suggestions for Authors

This study analyzed 1,052 proteins and 377 metabolites from the soleus muscle of rats after 28 days of hindlimb unloading (HU). Pathway analysis revealed slow-to-fast fiber type switching and changes in various metabolic pathways including glutathione, and ferroptosis pathways. While the findings provide some insights into muscle atrophy, the following comments should be addressed:

Major Comments

- Comparison with Other Models: Discuss the differences in differentially expressed proteins (DEPs) and metabolites (DEMs) between the HU model and other muscle atrophy models, such as myopathy and sarcopenia.

- Clinical insight as biomaker: Compare the DEPs and DEMs identified in this study with data from patients with muscle atrophy or similar conditions. This will help highlight differences and strengthen the authors' claims about potential biomarkers with clinical relevance.

- Fbp2 Interpretation: The proposed link between Fbp2-mediated mitochondrial dysfunction and fiber type switching requires stronger evidence. Incorporate findings from related studies on mitochondrial function in Fbp2 KO/KD models.

Pathway Interpretation: Provide a proposed pathway diagram or a clearer explanation to demonstrate how the DEPs and DEMs data presented in this study are pathophysiologically linked to HU-induced muscle atrophy. 

Minor Comments

- Abstract: Avoid unexplained abbreviations and repetitive content.

- Introduction: Include citations for statements lacking references.

- Methodology: Add details on how lean body mass was measured.

- Literature Reference: need a reference for ACTN's role in muscle fiber type switching (page 8).

Author Response

Response to Reviewer 1 Comments

1. Summary

Thank you very much for taking the time to review this manuscript. Please find the detailed responses below and the in the re-submitted files.

Major Comments

Comments 1: Comparison with Other Models: Discuss the differences in differentially expressed proteins (DEPs) and metabolites (DEMs) between the HU model and other muscle atrophy models, such as myopathy and sarcopenia.

Response 1:

Thank you for your insightful comments and suggestions on our manuscript.Hindlimb suspension is a commonly used model for simulating disuse muscle atrophy. In this model, the differential proteins (DEPs) and metabolites (DEMs) exhibit both similarities and unique characteristics when compared to other muscle atrophy models. As for DEPs, the downregulation of NDUFS4,NDUFS6L1,NDUFA7 after HU is consistent with previous findings of compromised mitochondrial respiration in glucocorticoid-induced muscle atrophy(Reference:Mitochondrial Dysfunction Launches Dexamethasone-Induced Skeletal Muscle Atrophy via AMPK/FOXO3 Signaling. Mol Pharm, 2016, 13). In this study, the DEMs observed share similarities with those found in the plasma of patients with sarcopenia, such as the elevation of carnitine and alterations in the histidine metabolism pathway(Reference: Novel metabolic and lipidomic biomarkers of sarcopenia.J Cachexia Sarcopenia Muscle, 2024 ,15). Interestingly, there is a discrepancy in terms of DEPs and DEMs between this study and the studies of muscle atrophy diseases, with some even showing an opposite trend. For example, the muscle proteome analysis of Facioscapulohumeral dystrophy patients reveals the decreased FBP2 whereas the upregulation of FBP2 was found in our study.[1] YAP abundance and activity in muscles is increased following injury or degeneration of motor nerves, as a process to mitigate neurogenic muscle atrophy. The increased YAP1, possibly as a subsequent biological response, promotes the protein synthesis in denervated skeletal muscles[2]. However, in our study, the downregulation of YAP1 after HU may be one of the initiating factors of disuse atrophy. In addition, some metabolic-related protein expression differences are reported for the first time in muscle atrophy, such as the expression of GLUL and glutathione transferase family members.

As for metabolites, due to the scarcity of current metabolomic studies on disuse muscle atrophy, many of the metabolites we report have not been previously documented.                  

However, the derangement of taurine metabolism induced by weightlessness has been reported to be related to Duchenne muscular dystrophy[3].We have already included differences in differentially expressed proteins (DEPs) and metabolites (DEMs) between the HU model and other muscle atrophy models in the discussion section. Please see Page 16 and 17.

Comments 2: Clinical insight as biomaker: Compare the DEPs and DEMs identified in this study with data from patients with muscle atrophy or similar conditions. This will help highlight differences and strengthen the authors' claims about potential biomarkers with clinical relevance.

Response 2:

We are grateful for this valuable suggestion which can make our research more meaningful. We have compared the DEPs and DEMs identified in this study with data from patients with similar conditions and found that FBP2 and ACTN3 were shown to be gravity-responsive muscle proteins[4]. The downregulation of NDUFS4,NDUFS6L1,NDUFA7 after HU is consistent with previous findings of compromised mitochondrial respiration in glucocorticoid-induced muscle atrophy[5]. Besides, the increased YAP1 is reported to promote the protein synthesis in denervated skeletal muscles[2]. In our study, the downregulation of YAP1 after HU may be one of the initiating factors of disuse atrophy.

In terms of DEMs, the disorders in histidine metabolism, arginine and proline metabolism in our study are also found in previous study under similar conditions[6]. 

Currently, there is a scarcity of studies on samples from patients with disuse muscle atrophy, with the majority being animal muscle unloading models. In our research, we compared our findings with those under similar conditions and identified FBP2, NDUFS4, YAP1, and ACTN3 as potential biomarkers for disuse muscle atrophy. However, the functions of these molecules in the process of muscle atrophy still require further investigation in both patients and animal models.

Fbp2 Interpretation: The proposed link between Fbp2-mediated mitochondrial dysfunction and fiber type switching requires stronger evidence. Incorporate findings from related studies on mitochondrial function in Fbp2 KO/KD models.

Pathway Interpretation: Provide a proposed pathway diagram or a clearer explanation to demonstrate how the DEPs and DEMs data presented in this study are pathophysiologically linked to HU-induced muscle atrophy.

Comments 3: Pathway Interpretation: Provide a proposed pathway diagram or a clearer explanation to demonstrate how the DEPs and DEMs data presented in this study are pathophysiologically linked to HU-induced muscle atrophy.

Response 3:

    Thank you for your request for further clarification on the pathway interpretation of how DEPs and DEMs data presented in our study are pathophysiologically linked to HU-induced muscle atrophy.In response to this important query, we have created a detailed flowchart that illustrates the intricate relationships and pathways connecting the DEPs and DEMs to the pathophysiology of HU-induced muscle atrophy. This visual representation has been incorporated into the manuscript and is designed to provide a clearer and more comprehensive understanding of the underlying mechanisms.

Minor Comments

Comments 1: Abstract: Avoid unexplained abbreviations and repetitive content.

Response 1:

Thank you for your guidance on improving the clarity and conciseness of our manuscript. We have taken your advice to avoid unexplained abbreviations and repetitive content seriously and have made the necessary revisions.

Please find the revised abstract in the updated manuscript, where we have addressed your concerns. We appreciate your feedback and are committed to maintaining the highest standards in our work

Comments 2: Introduction: Include citations for statements lacking references.

Response 2:

Thank you for your constructive feedback on our manuscript, particularly regarding the need for citations in the introduction section.

We have taken your advice into account and have now included the necessary citations for all statements that were previously lacking references. The updated manuscript now reflects these changes.

We appreciate your attention to detail and your commitment to ensuring the scholarly integrity of our research presentation.

Comments 3: Methodology: Add details on how lean body mass was measured.

Response 3:

    Thank you for your inquiry regarding the measurement details of lean body mass in our study. We appreciate your interest and would like to clarify that our study focused on measuring body mass, not lean body mass.

We chose to measure body mass as it provides a straightforward and comprehensive assessment of the rats' overall health and well-being. By monitoring changes in body mass, we were able to determine that the hindlimb unloading protocol did not lead to significant weight loss, suggesting that the rats maintained their health throughout the experimental period.

We hope this explanation addresses your question and provides the necessary context for the measurement approach taken in our study. We believe that the focus on body mass was appropriate for our research objectives and contributed to the overall understanding of the effects of hindlimb unloading on the rats.

Comments 4: Literature Reference: need a reference for ACTN's role in muscle fiber type switching (page 8).

Response 4:

Thank you for your request for a literature reference regarding the role of ACTN3 in muscle fiber type switching. We have added the necessary reference to our manuscript, which can be found on page 8 where the topic is discussed.

We appreciate your guidance and have ensured that our manuscript now includes the appropriate citation to support our statements.

Major revisions

Thank you for your insightful comment regarding the link between FBP2-mediated mitochondrial dysfunction and fiber type switching. We acknowledge the need for stronger evidence to support this relationship, and while we do not have direct evidence to demonstrate that FBP2 regulates the relationship between mitochondrial content and fiber type switching, we can infer a potential mechanism based on existing research.

Our hypothesis is that FBP2 may be involved in fiber type switching through the regulation of mitochondrial content. This is supported by the fact that FBP2 has been shown to inhibit mitochondrial biogenesis. Fast-twitch fibers are characterized by lower mitochondrial fusion rates and oxidative phosphorylation levels compared to slow-twitch fibers . Additionally, mitochondrial dysfunction has been implicated in promotion of muscle fibers from an oxidative (slow fiber) to a glycolytic (fast fiber) phenotype . In the hindlimb unloading (HU) model, FBP2 expression is significantly increased , suggesting a possible role in mediating fiber type switching in this context. Thus, We speculate that FBP2 may influence fiber type switching in the HU model by modulating mitochondrial dysfunction.

To address your suggestion, we have incorporated these findings into our manuscript to provide a clearer explanation of our hypothesis. We believe that these references strengthen our discussion and provide a foundation for the potential role of FBP2 in fiber type switching. Please see page 16.

References:

  1. Moriggi, M.; Ruggiero, L.; Torretta, E.; Zoppi, D.; Arosio, B.; Ferri, E.; Castegna, A.; Fiorillo, C.; Gelfi, C.; Capitanio, D. Muscle Proteome Analysis of Facioscapulohumeral Dystrophy Patients Reveals a Metabolic Rewiring Promoting Oxidative/Reductive Stress Contributing to the Loss  of Muscle Function. Antioxidants 2024, 13, doi:10.3390/antiox13111406.
  2. Watt, K.I.; Turner, B.J.; Hagg, A.; Zhang, X.; Davey, J.R.; Qian, H.; Beyer, C.; Winbanks, C.E.; Harvey, K.F.; Gregorevic, P. The Hippo pathway effector YAP is a critical regulator of skeletal muscle fibre size. Nat. Commun. 2015, 6, 6048, doi:10.1038/ncomms7048.
  3. Merckx, C.; De Paepe, B. The Role of Taurine in Skeletal Muscle Functioning and Its Potential as a Supportive Treatment for Duchenne Muscular Dystrophy. Metabolites 2022, 12, doi:10.3390/metabo12020193.
  4. Ino, Y.; Ohira, T.; Kumagai, K.; Nakai, Y.; Akiyama, T.; Moriyama, K.; Takeda, Y.; Saito, T.; Ryo, A.; Inaba, Y. et al. Identification of mouse soleus muscle proteins altered in response to changes in gravity loading. Sci. Rep. 2023, 13, 15768, doi:10.1038/s41598-023-42875-8.
  5. Liu, J.; Peng, Y.; Wang, X.; Fan, Y.; Qin, C.; Shi, L.; Tang, Y.; Cao, K.; Li, H.; Long, J. et al. Mitochondrial Dysfunction Launches Dexamethasone-Induced Skeletal Muscle Atrophy via AMPK/FOXO3 Signaling. Mol. Pharm. 2016, 13, 73-84, doi:10.1021/acs.molpharmaceut.5b00516.
  6. Zhou, Y.; Liu, X.; Qi, Z.; Yang, L.; Huang, C.; Lin, D. Deciphering the Therapeutic Role of Lactate in Combating Disuse-Induced Muscle Atrophy: An NMR-Based Metabolomic Study in Mice. Molecules 2024, 29, doi:10.3390/molecules29102216.

Reviewer 2 Report

Comments and Suggestions for Authors

Comments to the Authors

Mechanical unloading-induced skeletal muscle atrophy leads to reduced physical strength and an increased risk of disability. Understanding the underlying mechanisms of this condition is crucial for developing effective interventions. The authors Yuan Wanga et al. present a work in which they performed for the first time, integrated proteomic and metabolomic analysis of muscle atrophy induced by hindlimb unloading.

The manuscript is well-structured, logically presented, and supported by substantial statistically significant experimental data, complemented by comparative analysis with existing literature. There are no major technical or conceptual issues that would preclude its publication.

However, I recommend the following edits to enhance the clarity and accessibility of the manuscript for a broader readership:

1.       Detailed Description of TMT Labeling: In section 3.2, the authors briefly mention using the tandem mass tag (TMT) mass spectrometry method for quantification: “We used the tandem mass tag mass spectrometry method to analyze soleus tissue…”. This procedure should be described in greater detail, particularly in the Discussion and Materials and Methods sections. Including more specifics about the TMT labeling workflow, as illustrated in similar studies1, 2, would provide better context for readers unfamiliar with this technique.

2.       Violin Plots in Figures 3A and 4A: To improve clarity, I suggest enlarging the panels containing violin plots in Figures 3A and 4A. Additionally, reorganize the plots to create a more structured layout, clearly indicating whether the observed changes represent up-regulation or down-regulation.

3.       Optimization of Figures 5D and 5F: Enhance the visibility of details and adjust the font sizes in Figures 5D and 5F to ensure all elements are legible, even at smaller scales. This will help readers more easily interpret the data presented.

REFERENCES

1.            Thompson, A.;  Schafer, J.;  Kuhn, K.;  Kienle, S.;  Schwarz, J.;  Schmidt, G.;  Neumann, T.; Hamon, C., Tandem mass tags: A novel quantification strategy for comparative analysis of complex protein mixtures by MS/MS. Analytical Chemistry 2003, 75 (8), 1895-1904.

2.            Stepath, M.;  Zulch, B.;  Maghnouj, A.;  Schork, K.;  Turewicz, M.;  Eisenacher, M.;  Hahn, S.;  Sitek, B.; Bracht, T., Systematic Comparison of Label-Free, SILAC, and TMT Techniques to Study Early Adaption toward Inhibition of EGFR Signaling in the Colorectal Cancer Cell Line DiFi. J Proteome Res 2020, 19 (2), 926-937.

Author Response

Response to Reviewer 2 Comments

1. Summary

Thank you very much for taking the time to review this manuscript. Please find the detailed responses below and the in the re-submitted files.

2. Minor comments:

Comments 1: Detailed Description of TMT Labeling: In section 3.2, the authors briefly mention using the tandem mass tag (TMT) mass spectrometry method for quantification: “We used the tandem mass tag mass spectrometry method to analyze soleus tissue…”. This procedure should be described in greater detail, particularly in the Discussion and Materials and Methods sections. Including more specifics about the TMT labeling workflow, as illustrated in similar studies1, 2, would provide better context for readers unfamiliar with this technique.

Response 1:

Thank you for your meticulous review and for pointing out the discrepancy in our manuscript regarding the mass spectrometry method used. We sincerely apologize for the oversight and appreciate your attention to detail.

Upon your feedback, we have identified and corrected the error in section 3.2. Actually, we did not use the tandem mass tag (TMT) mass spectrometry method as initially mentioned. Instead, we utilized the LC-MS/MS method for our proteomic analysis.

We have made the necessary revisions to the manuscript to accurately reflect the methodology used. The Materials and Methods section has been updated to detail the LC-MS/MS procedure. The corrections are presented in section2.4 and section 3.2. We have ensured that the description of our methodology is now consistent throughout the manuscript.

We apologize for any confusion caused by the initial error and thank you for your understanding. We appreciate your understanding and patience in this matter and are grateful for the opportunity to improve the quality of our manuscript. We have taken measures to ensure that such errors do not occur in future submissions.

Comments 2: Violin Plots in Figures 3A and 4A: To improve clarity, I suggest enlarging the panels containing violin plots in Figures 3A and 4A. Additionally, reorganize the plots to create a more structured layout, clearly indicating whether the observed changes represent up-regulation or down-regulation.

Response 2:

Thank you for your valuable feedback and suggestions regarding the presentation of our data in Figures 3A and 4A.

We have taken your comments into consideration and have made the necessary revisions to enhance the clarity and readability of our figures. Specifically, we have enlarged the panels containing the violin plots in Figures 3A and 4A to improve the overall clarity of the presentation. Additionally, we have reorganized the plots to create a more structured layout, which now clearly distinguishes between up-regulated and down-regulated differentially expressed proteins.

We believe that these revisions, as detailed in the updated Figures 3 and 4, will significantly enhance the reader's ability to interpret the data and grasp the significance of the observed changes in protein expression. We appreciate your guidance and are confident that these modifications will significantly improve the quality and comprehension of our manuscript.

Comments 3: Optimization of Figures 5D and 5F: Enhance the visibility of details and adjust the font sizes in Figures 5D and 5F to ensure all elements are legible, even at smaller scales. This will help readers more easily interpret the data presented.

Response 3:

    Thank you for your helpful suggestions regarding the optimization of Figures 5D and 5F in our manuscript.

We have taken your feedback to heart and have made the necessary adjustments to enhance the visibility of details within these figures. Specifically, we have increased the image size and resolution to ensure that all elements are legible. This improvement will greatly facilitate the readers' ability to interpret the data presented.

The revised Figures 5D and 5F, with the enhanced quality, are included in the revised manuscript. We believe that these changes will make the figures more accessible and the information more digestible for our audience.

We appreciate your attention to detail and your commitment to improving the clarity of our presentation.